# HMGA Proteins in Hematological Malignancies

**DOI:** 10.3390/cancers12061456

**Published:** 2020-06-03

**Authors:** Angela Minervini, Nicoletta Coccaro, Luisa Anelli, Antonella Zagaria, Giorgina Specchia, Francesco Albano

**Affiliations:** Department of Emergency and Organ Transplantation (D.E.T.O.), Hematology Section, University of Bari, 70124 Bari, Italy; minervini.angela@gmail.com (A.M.); nicoletta.coccaro@uniba.it (N.C.); luisa.anelli@uniba.it (L.A.); antonellazagaria@hotmail.com (A.Z.); specchiagiorgina@gmail.com (G.S.)

**Keywords:** HMGA proteins, hematologic malignancies, molecular markers, targeted therapy

## Abstract

The high mobility group AT-Hook (HMGA) proteins are a family of nonhistone chromatin remodeling proteins known as “architectural transcriptional factors”. By binding the minor groove of AT-rich DNA sequences, they interact with the transcription apparatus, altering the chromatin modeling and regulating gene expression by either enhancing or suppressing the binding of the more usual transcriptional activators and repressors, although they do not themselves have any transcriptional activity. Their involvement in both benign and malignant neoplasias is well-known and supported by a large volume of studies. In this review, we focus on the role of the HMGA proteins in hematological malignancies, exploring the mechanisms through which they enhance neoplastic transformation and how this knowledge could be exploited to devise tailored therapeutic strategies.

## 1. Introduction

The high mobility group AT-Hook (HMGA) proteins are a family of nonhistone chromatin remodeling proteins regulating gene expression, known as “architectural transcriptional factors” [1,2,3,4]. The gene family comprises the *HMGA1* and *HMGA2* genes; the former is located on chromosome 6 (6p21), is about 10 kb large, and consists of eight exons [5]; the latter is located on chromosome 12 (12q14-15), is ≥160 kb in size, and consists of five exons [6]. Alternative splicing of the *HMGA1* gene produces three messenger RNA (mRNA) and three corresponding proteins, HMGA1a (formerly HMG-I), HMGA1b (formerly HMG-Y), and HMGA1c (formerly HMG-I/R) [7]. The HMGA1b isoform differs from the 1a isoform in that it lacks 11 amino acids; instead, the HMGA1c isoform is produced by alternative splicing of the *HMGA1* gene using noncanonical acceptor and donor sites. As regards HMGA2, recently, two isoforms (HMGA2-L and HMGA2-S) have been described for which expression seems to be confined to the hematopoietic compartment and to depend on the stage of development of the hematopoietic stem cell (HSC) (see below) [8].

Sequence alignment showed that HMGA2 has a homology of about 55% with HMGA1 (A and B isoforms), including AT-hooks that are located in separate exons [6]. The AT-hook motif is positively charged and consists of a stretch of nine amino acids, containing the Arg-Gly-Arg-Pro repetition. This motif allows the protein to recognize and bind the AT-rich sequences in the minor groove of B-form DNA, rendering it able to recognize a structure rather than a specific sequence [6,9], so that some data report the ability of the HMGA1 protein to bind DNA even in regions without a rich AT content [10,11,12]. Once bound to DNA, the AT-hook undergoes a conformational change [13], and simultaneous binding of two or more AT-hooks to different binding sites leads to an increase in the strength of HMGA interactions with DNA [14]. 

By binding the minor groove of AT-rich DNA sequences, HMGA proteins interact with the transcription apparatus, altering the chromatin modeling and regulating gene expression by either enhancing or suppressing the binding of the more usual transcriptional activators and repressors, although they do not themselves have any transcriptional activity [2,6,15,16,17,18]. Their function is critical for the formation of higher-order nucleoprotein complexes (called enhanceosomes) in the promoter region, allowing the efficient transcriptional activation of a gene [19] thanks to their multiple surfaces capable of protein–protein interactions [6,13]. Through such interactions, HMGA proteins may directly or indirectly control transcription, inducing conformational changes in chromatin or in the transcription factors themselves. In such ways, HMGA proteins are able to influence a wide variety of normal biological processes, such as cell growth, proliferation, differentiation, and death [2,6]. They show an active role in the division cycle, and data indicate that both histone H1 and HMGA1 are necessary for chromatin condensation [20,21]. 

HMGA1 and 2 display different target genes; for example, the *BRCA1* gene is negatively regulated by HMGA1 but not by HMGA2 [22]. Instead, HMGA2 has several other targets, including cycle regulators, one of which is the *CDKN2A* gene, encoding the two proteins p16INK4A and p14ARF. With the RNA-binding proteins LIN28A/B and the microRNA (miRNA) *let-7b*, HMGA2 and CDKN2A form an axis that controls stem cell aging in both normal and pathological conditions [23,24,25]. 

## 2. HMGA Proteins in Oncology

HMGA expression seems to be ubiquitous and high during embryogenesis and null or hardly detectable in adult differentiated tissues [1,2,3,4,26,27,28,29,30]. Several studies demonstrated the involvement of these proteins in embryonic development [31], adipocytes cell growth, and differentiation. Mice lacking the *HMGA2* gene showed a pygmy phenotype with a drastic reduction of fat tissue [32]; instead, HMGA2 overexpression results in giant, obese mice [33]. On the contrary, HMGA1 inhibits adipocytes cell growth and triggers differentiation [34].

Overexpression of both genes is observed in most human malignant neoplasias, linking their alteration to carcinogenesis [2,6,35,36,37,38,39], while blocking their expression prevents thyroid cell transformation and triggers the death of malignant cells [40,41,42]. Various in vitro and in vivo studies have demonstrated the oncogenic potential of HMGA proteins; their overexpression transforms mouse and rat fibroblasts [43,44], and in transgenic mice, the development of natural killer T-cell lymphomas and pituitary adenomas has been observed [45,46,47,48]. There are high expression levels of *HMGA* transcripts in embryonic stem (ES) cells [49,50], HSCs [50,51,52,53], leukemic stem cells [30], and poorly differentiated or refractory tumors [1,2,3,4,30,44,46,51,52,54,55,56,57,58,59,60,61,62]. In cultured human lymphoid cells, the ectopic expression of HMGA1a triggers leukemic transformation [44,62] and, in transgenic mice, the development of an aggressive T-cell lymphoid neoplasia [46,47,58,63,64]. In particular, *HMGA1* is overexpressed in several high-grade or refractory neoplasias, including hematologic malignancies and solid tumors [3,46,52,58,60,62,63,65]. Indeed, HMGA1 proteins are the most abundant nonhistone, chromatin-binding proteins in tumor cells [3].

Chromosomal duplications or translocations cause HMGA1 overexpression in rare cases; more frequently, the translocations involve *HMGA2* (chromosomal region 12q14-15) in benign tumors, such as lipomas [66] and uterine fibroids [67,68,69]. With Fluorescent in Situ Hybridization (FISH) analysis, it has been observed that, in most cases, the breakpoint in the *HMGA2* locus involves exons 1-2 or 1-3, leading to the loss of the C-terminal tail [70], separating the DNA-binding domains from the acid tails, and merging them with ectopic sequences. This seems to be sufficient for tumor development. Further studies revealed the presence of multiple binding sites for miRNA *let-7* in the 3′-Untranslated Region (3′-UTR) of *HMGA2* [71]. This evidence corroborates the observation that, in tumors with rearrangements destroying 3′-UTR of *HMGA2*, there were increased levels of the intact protein [68,72,73]; this suggests the presence of functional sites for the binding of miRNAs (such as *let-7*, *mir-98*, and *miR-33*) [74,75,76,77,78,79,80,81,82,83,84,85,86] that are important in regulating the gene expression.

A large volume of data demonstrates that miRNA dysregulation may intervene in tumor development [87,88]. Cancers exhibit distinctive miRNA expression signatures, raising the speculation that a dysregulation of specific pathways intervenes in tumorigenesis mechanisms in a specific tissue [89,90,91]. 

The repression of *let-7* is necessary to enhance cell proliferation [74,75,79]; in fact, *let-7* expression is inversely related to the expression of HMGA2. This finding supports the hypothesis that *let-7* can act as a repressor of HMGA2 [6,92]. 

## 3. HMGA Proteins in Hematopoiesis

In the hematologic field, HMGA proteins seem to play a pivotal role in both physiological and pathogenic conditions.

The loss of HMGA2 in HSCs does not allow a correct fetal hematopoietic process, causing slower self-renewal and proliferation rates, while it seems not to be indispensable in adult hematopoiesis [93]. Indeed, in 2013, in mouse models, Copley et al. demonstrated that the fine equilibrium between levels of the RNA-binding protein LIN28B and *let-7* strikingly influence HSC developmentally timed changes, observing that HSC self-renewal potentials reduced when LIN28B decreased and *let-7* increased [93]. As *HMGA2* expression is high during embryogenesis, being particularly upregulated in fetal HSCs and gradually decreasing during the transition to adult hematopoiesis, experiments in transplanted irradiated hosts to define the role of HMGA2 and LIN28 in HSC functions showed that both LIN28- and HMGA2-induced overexpression increased the self-renewal activity of adult HSCs. However, HMGA2 overexpression in adult HSCs did not seem sufficient to mimic all the effects of elevated LIN28B that triggers a fetal lymphoid differentiation program. Accordingly, *HMGA2* seems to be a downstream modulator of the HSC self-renewal ability; its function is regulated by LIN28B, that acts as a master regulator of developmentally timed changes of HSC [93]. However, this regulatory mechanism seems to be even finer tuned [91]. In a recent work, Cesana et al. employed high-throughput genomic approaches to profile miRNAs, long intergenic non coding RNAs (lincRNAs), and mRNAs in HSCs during development in order to characterize transcriptional and posttranscriptional changes [8]. These analyses highlighted distinct alternative splicing patterns of HSCs in various key hematopoietic regulators, one of which is HMGA2, for which an alternative isoform, escaping miRNA-mediated control, was identified. It seems that the splicing kinase CLK3 conserves HMGA2 function in response to an increase of *let-7* miRNA levels by regulating HMGA2 splicing. These data delineate a CLK3/HMGA2 functional axis regulating HSC functions during development [8]. In humans, *HMGA2* presents alternative splicing isoforms, *HMGA2-L*, which is sensitive to degradation by *let-7*, being the dominant isoform in fetal HSCs, whereas newborn HSCs express the *HMGA2-S* isoform, that is resistant to degradation by *let-7* [8] (Figure 1). In support of the crucial role of HMGA2 in regulating HSCs properties, it has been found in gene therapy experiments that a deliberate overexpression of *HMGA2*, in parallel with that of the transgene of interest, might be needed to obtain the therapeutic effect [94,95], particularly in cases where a large fraction of corrected HSCs must be obtained to gain the benefit. In the case reported in 2010 by Cavazzana et al. of an adult patient with severe βE/β0-thalassaemia treated with gene therapy, the presence of a dominant, myeloid-biased cell clone was demonstrated, in which the integrated vector transferring the β-globin gene caused the transcriptional activation of *HMGA2* in erythroid cells and further increased the expression of a truncated *HMGA2* mRNA insensitive to degradation by *let-7* miRNAs [94]. 

There is evidence in mice that Hmga2 is also a regulator of gamma-globin mRNA, and that it moderately increased HbF levels in human adult erythroblasts in vitro [96]. Targeted inhibition of *let-7* is sufficient for specific developmental changes to occur in gamma-globin transcription and HbF levels through increased *HMGA2* levels [97].

As regards the LIN28B-*let7*-HMGA2 axis, it is also involved in the megakaryocyte and platelet lineage. LIN28B expression in fetal myelo-erythroid progenitors suggests that it could have a role in the prenatal platelet-forming lineages; indeed, megakaryocytes derived from human ES cells express higher levels of LIN28B as compared to adult controls [98]. This implies that LIN28B may also have an active function in megakaryocyte development and thus platelet function [99]. 

In vitro experiments on ES cells lacking one or both *HMGA1* alleles showed that these cells showed a lesser differentiation in T-cell precursors than Wild Type (WT) ES cells and preferentially in B cells, maybe because of an increased interleukin 6 and decreased interleukin 2 expression [100]. Furthermore, there was evidence of an aberrant hemopoietic differentiation, with a reduction of the monocyte/macrophage population and an increase in megakaryocyte precursors, erythropoiesis, and globin gene expression. The restoration of *HMGA1* expression reestablished the WT phenotype. These studies highlighted that HMGA1 proteins directly control the transcription of GATA-1, a key transcription factor which regulates red blood cell differentiation, that is overexpressed in *HMGA1*−/− ES cells. This could offer an explanation, at least in part, of the alteration of the lineages of megakaryocytes/erythrocytes [100]. These observations add weight to those of Pierantoni et al., who observed that the induction of HMGA1 protein synthesis appears to play a role in the differentiation of megakaryocytes while its inhibition may be necessary in the differentiation of erythrocytes or macrophages [101].

Regarding lymphoid compartment, animal experiments showed HMGA2 involvement in mouse thymopoiesis, as it is highly expressed in fetal and neonatal early T cell progenitors (ETP), the most intrathymic precursors, while it results almost undetectable 5 weeks after birth [102]. In the same way, the number of thymic cells increases during the first two weeks after birth, stabilizes, and then declines by seven weeks of age. In fact, HMGA2-deficient mice showed deficient ETPs after birth, and also the total thymocyte number was fewer as compared to WT. Bearing in mind that, also in human fetal thymic progenitors, HMGA2 expression is high and decreases during growth, it is possible that a similar mechanism takes place [102].

As regards HMGA1, studies using HMGA1-green fluorescent protein fusion explored HMGA1 expression in undifferentiated and differentiated populations of hematopoietic cells, demonstrating that HMGA1 is highly expressed in CD4/CD8-double negative (DN) cells and transiently downregulated in CD4/CD8-double positive (DP) cells during early T cell development in the thymus [103]. Indeed, HMGA1 binds directly to cis-regulatory elements in the *CD4/CD8* loci in DN cells but not in DP cells. Moreover, in DN leukemic cells, CD4/CD8 expression is induced by the inhibition of HMGA1, and T leukemic cells lacking HMGA1 showed a markedly decreased proliferation [103]. 

HMGA1 indirectly controls *µ* enhancer activity during B lymphocyte development, acting on the combinatorial activity of transcription factors [104,105,106,107]. Various studies revealed that HMGA1 interacts with both PU.1 and ETS proto-oncogene 1 transcription factor 1 (Ets-1) in solution [108,109]; HMGA1 interaction with PU.1 likely induces a change in PU.1 structure, increasing PU.1/*µ* enhancer binding. This augmented binding renders HMGA1 able to boost PU.1/Ets-1 functional synergy on the *µ* enhancer [108,110].

In view of the role of HMGA proteins in driving both benign and malignant tumors [2] and the evidence regarding their main functions in various hematopoietic physiological mechanisms, various research studies have explored their involvement in hematological malignancies.

## 4. HMGAs Involvement in Myeloid Malignancies

### 4.1. HMGAs Involvement in Myeloproliferative Neoplasms

Myeloproliferative neoplasms (MPN) are a group of clonal diseases of the hematopoietic system which feature an excessive production of myeloid cells. This group includes polycythemia vera (PV) essential thrombocythemia (ET), primary myelofibrosis (PMF), and chronic myeloid leukemia (CML) [111]. These four diseases share clinical features [112], but from a biological point of view, it is possible to subdivide them into Philadelphia chromosome (Ph) positive (Ph+) (CML) and Ph negative (Ph-) (PV, ET, and PMF) diseases. CML is characterized by the Ph [113], derived from the balanced translocation of chromosome 9 and chromosome 22 [114]; driver mutations of *JAK2*, *MPL*, and *CALR* together account for approximately 90% of Ph-MPNs [115]. The most frequent mutation is the point mutation of the Janus kinase 2 (*JAK2*) gene (*JAK2V617F*) that can be observed in most patients with PV (95%), ET (50%), and PMF (60%) [116,117,118,119]. The *JAK2V617F* involves the constitutive activation of the JAK-STAT signal pathway which confers a blood cells growth advantage. It has been hypothesized that the cooperation of different genetic accidents is necessary to allow the initiation and progression of the disease; a mutation alone is not sufficient for the development of the disease in humans [120]. In some cases of MPN, an overexpression of *HMGA2* provides an important contribution to the mechanisms of cell proliferation and differentiation [70,121,122,123,124,125]. The first group to relate the upregulation of HMGA2 and MPN was Guglielmelli et al.; studying the molecular profile in PMF of CD34+ cells, they observed the presence of the *JAK2V617F* mutation that influenced the abnormal expression of *HMGA2* [125].

As stated above, the *HMGA2* gene possesses, at the level of the 3′-UTR, seven complementary sequences to *let-7* which negatively regulate the protein expression [126]. Deletion or truncation of *HMGA2* 3′-UTR determines the loss of the *let-7*-binding sites, which leads to overexpression of the full-length or truncated protein with a conserved DNA binding capacity and a high likelihood of promoting tumors [127,128]. In their 2011 study, Ikeda et al. [129] created a *HMGA* transgenic mouse with 3′-UTR deletion and therefore overexpression of the *HMGA2* mRNA (Δ*HMGA2* mouse); a similar genetic alteration had already been observed in MPN [70,73,121,122,123,124,125]. Their data showed a clinical scenario similar to that of MPN, with an increase of all blood cells, splenomegaly, hypercellularity of the bone marrow (BM), an increased formation, and growth of erythroid colonies, independently of erythropoietin. From a biological point of view, the Δ*HMGA2* mice showed an increased expression of *JAK2* mRNA and phosphorylation of STAT3 and AKT, independently of the cytokines. Therefore, overexpression of *HMGA2* through the constitutive JAK-STAT pathways and AKT activation may provide a proliferative advantage to the hematopoietic stem and progenitor cells [129].

Experiments with Δ*HMGA2* mice have shown that the expression of HMGA2 promotes clonal growth in MPNs by improving the self-renewal and functionality of HSC [129] as well as other types of stem cells [24,130,131]. Further data confirming this hypothesis were provided by the Ueda et al. group, that created a Δ*HMGA2/JAK2V617F* transgenic mouse which overexpresses *HMGA2* following the deletion of the 3′-UTR. Compared to JAK2V617F mice, these mice showed a more serious leukocytosis, splenomegaly, and anemia with reduced survival; on the other hand, the severity of the myelofibrosis was comparable. In BM serial transplantation experiments, Δ*HMGA2*/JAK2V617F cells conferred a greater repopulation capacity that emulated a severe MPN, as compared to JAK2V617F cells; this establishes that HMGA2 is able to promote the progression of MPN at the HSC level [132].

The work by Chen et al. in 2017 emphasized a survival advantage conferred by the JAK2 mutation, given that apoptosis is inhibited in these cells. Patients with *HMGA2* overexpression are more likely to develop ET and to experience thrombotic episodes [115]. Numerous other studies have highlighted the increased platelet count in this group of patients and the possibility that they may belong to the ET subtype; already in 2012, Oguro et al. had observed that, in HSC, the overexpression of HMGA2 triggered a myeloproliferative condition in mice, with increased megakaryopoiesis [133], and in 2016, Yang et al. observed the same in mice with JAK2 mutation [134]. While BM cells that upregulated HMGA2 have a proliferative advantage over control competitive repopulation assay in mice and BM transplant models [129], it is possible that overexpressed *HMGA2* triggers a hematopoietic proliferation mechanism with a preference for platelets and inhibits the apoptosis mechanism [133,134]. All these works have raised the hypothesis that alteration of the HMGA2 protein could confer a proliferative advantage to altered progenitors, triggering the pathogenesis of MPN or other clonal hematological diseases [127], but how HMGA2 intervenes directly in the pathogenesis of MPN remains to be clarified. 

The overexpression of *HMGA2* associated with the 3′-UTR deletion was found not only in MPN patients but also in those with myelodysplastic syndrome (MDS) and MDS/MPN [70,121,122,123,124,125] as well as in patients with paroxysmal nocturnal hemoglobinuria (PNH) [73] (Figure 2). PNH is characterized by the absence on the cell surface of the glycosyl phosphatidylinositol protein on the stem cell. All cell lines deriving from this stem cell show the same alteration and an increased expression of *HMGA2* due to the truncation of the 3′-UTR [73,127]. 

Although several studies have confirmed that the alteration involving chromosome 12 leads to the overexpression of *HMGA2* in patients with MPN [122,135], data from the work by Chen et al. demonstrated that the regulation of *let-7a* miRNA has a very important impact on the expression of HMGA2 in patients with MPN. These data are in line with a paper reporting that a reduced expression of *let-7* miRNA is the main cause of the alteration of *HMGA2* mRNA expression in MPN [36]. Meyer et al. observed an increase in the level of HMGA2 in the accelerated phase and in the blast crisis of CML compared to the chronic phase; furthermore, the expression of the protein is inversely correlated with *let-7* [136,137]. Several works have correlated the altered miRNA expression with the genetic complexity of MPNs, including *let-7a*, *miR-150*, *let-7g*, *let-7f*, *miR_4319*, and *miR-149* [138,139,140,141]. 

By gene expression profiling of the granulocytes of patients with PV, Bruchova et al. observed that the frequency of the JAK2 mutation is inversely proportional to the expression of *let-7a*. However, the overexpression of *HMGA2* has been observed in patients with PMF but not in patients with PV, among which a correlation between the expression levels of *HMGA2* and *let-7a* was not observed [138]; most likely, this figure was influenced by the small group of patients enrolled in the study [115]. Although an inverse relationship between the expression of *HMGA2* and *let-7a* has been reported, Harada-Shirado et al. failed to observe any possible relationship between the *JAK2V617F* mutation and the expression of *HMGA2* [36]. Furthermore, the upregulation of HMGA2 has also been reported in some MPN patients without the *JAK2* mutation [142]. Chen et al. also highlighted that patients who have increased *HMGA2* expression are more likely to have one of the MPNs driver mutations [115]. However, there are still too few data available in these papers to be able to draw conclusions. The interaction between *HMGA2* and *JAK2V617F* as well as whether, in patients with *JAK2*-mutated MPN, *HMGA2* overexpression has a specific role in the disease pathogenesis have yet to be clarified [115].

The improvement on the HSC properties due to the expression of HMGA2 has also been observed in some human gene therapy studies using lenti-viral or retro-viral for the transduction of human transgenes [94,95]. In some cases, the virus containing the gene of interest integrated into the genome at the level of the HMGA2 locus, consequently causing the loss of binding sites for the *let-7* miRNA and the clonal expansion of the hematopoietic cells. This intervention leads to a long-term effect, which in some patients with severe thalassemia results in independence from continuous blood transfusions [94]. 

The increased expression or/and truncation of *HMGA2* has been observed in patients with both MDS and PNH, in which the common feature is marrow failure [70,73,143]; the failure is in part due to the immunological injury of HSC orchestrated by the self-reactive cytotoxic T lymphocytes (CTL) that secrete the Tumor Necrosis Factor-α (TNF-α) and Interferon-γ (INF-γ) [144,145,146]. It has been hypothesized that an altered hematopoietic clone may survive the attack of IFN-γ produced by CTLs in these pathologies [147], suggesting that a genetic event following the HSC lesion is necessary until the altered cell acquires a clonal growth advantage and can expand (two hit-hypothesis) [148]. As already observed in MDS and PNH, it has recently been shown that TNF-α can induce clonal selection of JAK2V617F+ cells in MPN patients, probably because the action of TNF-α collides with the survival advantage of these cells [149]. It is clear that the pathogenesis of MPN is linked to the orchestrated action of genetic anomalies and immunological or humoral mechanisms. As evidence of this, the use of JAK2 inhibitors shows benefits in patients with PMF by reducing the size of the spleen and in part by reducing the concentration of cytokines [150]. At present, there is no clear information about the correlation between HMGA2 and cytokine production, although the microarray analysis performed in the 2011 study showed the activation of the immune response-related pathway in HSC of Δ*HMGA2* mice [129]. 

Therefore, HMGA2 could be an excellent candidate as a therapeutic target because it is involved at various levels in the pathogenesis of MPN, including the regulation of gene expressions and the hematopoietic clonal proliferation [127].

### 4.2. HMGAs Involvement in Acute Myeloid Leukemia

Acute myeloid leukemia (AML) is one of the most common blood cancers to affect adults, accounting for approximately 3% of all cancer cases and approximately 25% of all leukemias [151].

AML is a heterogeneous neoplasm, in which clonal proliferation of HSCs is observed, characterized by a blocked or severely compromised differentiation process and a progressive accumulation of blasts at various stages of maturation at the level of the BM, peripheral blood (PB), and other tissues. All this translates into the destruction of the hematopoietic system [151,152]. The prognosis for AML is severe, and the 5-year survival rate is estimated to be around 20%. The etiology of the disease is very complex, and there are many risk factors. Several studies have highlighted recurrent genetic and chromosomal alterations that may underlie the mechanisms of the disease [153,154,155]. Different molecular biology techniques are used to identify the most common genetic alterations, such as *NPM1, FLT3, IDH, CEBPA, RUNX1, ASXL1*, and *TP53*. The identification of the presence/absence of these genetic alterations and the cytogenetic data allow not only to perform the risk stratification and to better define the prognosis but also to facilitate the best therapeutic choice [156,157,158,159,160,161]. 

A new biomarker has recently been identified that has made an important contribution to the prognostic stratification of patients with AML. An increased expression of *HMGA2* is found in approximately 22% of patients and in 60% of patients with a mutation related to an unfavorable prognosis. High expression levels of *HMGA2* in patients with AML are related to a lower frequency of complete remission (about 59% vs. 83%), relapse-free survival (about 11% vs. 44%), and a poor 3-year overall survival (OS) (13.2% vs. 43%) [162]. 

Nyquist et al. [163] showed that t(12;13)(q14;q31) is associated with an upregulation of HMGA2 in AMLs, but the role played by the altered *HMGA2* expression in the AML mechanism remains unknown. In a 2016 study, Tan et al. aimed to define this relationship by studying the expression of the gene and its biological function in the blasts of patients with de novo AML and leukemic cell lines. Comparing mRNAs levels of HMGA2, they observed significantly higher amounts in the patient samples and cell lines than in normal cells; they also noted that the majority of patients who did not achieve remission had increased *HMGA2* mRNA and protein expression [164]. 

There are several signaling pathways that are known to be protagonists in the progression of leukemias, such as the PI3K/Akt/mTOR, Wnt/β-catenin, JAK/STAT, and NF-κB pathways [165,166,167,168]. The PI3K/Akt/mTOR signal pathway appears to play an important role in the proliferation effect mediated by HMGA2 [164]; this signal pathway is constitutively active in various types of cancer and is crucial in regulating the growth of malignant cells [169]. It is known that the PI3K/Akt/mTOR pathway plays a fundamental role in a vast number of different physiological processes such as cell cycle progression, transcription, translation, metabolism, differentiation, and apoptosis, all of which are reflected on growth, proliferation, survival, and tumorigenesis [170]. The altered expression of *HMGA2* in AMLs is coherent with the already known oncogenic role of mTOR and Akt in leukemias. In fact, data show that the reduced expression of *HMGA2* inhibits the PI3K/Akt signaling pathway and therefore inhibits cell proliferation through the downregulation of mTOR expression. It is therefore deduced that the PI3K/Akt/mTOR pathway is closely associated with the expression of *HMGA2* [164]. Tan et al. demonstrated, in vivo and in vitro, the oncogenic role of HMGA2 in AML since its downregulation hinders proliferation in AML cell lines [164]. 

Several studies have revealed the involvement of HMGA2 in different signaling pathways [9,164,171], but Yang et al. [172] were the first to observe the effect of HMGA2 expression on the WNT/β-catenin pathway in AML, while this effect had already been observed in other types of tumors [173,174]. Yang at al. used plasmids that overexpressed or silenced *HMGA2* in different cell lines, showing that silencing of HMGA2 inhibits Wnt and the levels of active β-catenin, reducing cell viability, invasion, and migration and improving cell apoptosis. The overexpression of *HMGA2* elicited inverse results. Furthermore, in this scenario, the X-linked inhibitor of apoptosis and Bcl-2 mRNA and protein levels are mitigated while Bax and cleaved-caspase-3 are overexpressed [172]. 

Daunorubicin (DNR) is one of the chemotherapy drugs used in the first line. DNR is an anthracycline-based chemotherapy with potent antitumor characteristics [175], but some patients exhibit drug resistance [176]. Yang at al. showed that inhibition of AML cells by DNR improved with HMGA2 silencing and that overexpression causes an adverse effect [172].

The expression of *HMGA2* was observed in CD34 + stem cells from healthy donors and in PB from AML patients, while it was not observed in normal blood samples. The high expression of *HMGA2* is related to the non-differentiated state of the leukemic cells [136,177]. Furthermore, there are data showing that the altered expression of *HMGA2* has a main role in malignant evolution by directly inducing the transformation of myeloid cells [64]. Several papers have fueled the suspicion that the deregulation of *HMGA2* expression is sufficient to interfere with the differentiation mechanism of cells in AML [178,179]. Indeed, in experiments involving *HMGA2* silencing, the induction of the monocytic-granulocytic differentiation mechanism in leukemic cells was observed. Although this correlation has been observed [171], the mechanism through which it occurs has not yet been clarified. 

There are data showing that *Homeobox A9 (HOXA9)* downregulation is fundamental for normal myeloid differentiation [180] (Figure 3A); moreover, its overexpression is able to preserve the self-renewal capacity of leukemic stem cells and to block the differentiation processes leading to leukemogenesis. Different authors have explored the possible role of *HOXA9* in the pathogenesis of AML [178,179]. Furthermore, in a 2013 work, it was observed that *HMGA2* can regulate *HOXA9* in the growth stage of breast cancer and the formation of metastases [181]. In about 70% of patients with AML, HOXA9 is overexpressed, probably due to the important differentiation block that occurs in leukemic cells [182,183]. The Tan group, by means of experiments on cell lines, observed an inhibition of *HOXA9* expression in *HMGA2* knockdown cells, while the downregulation of *HOXA9* supports differentiation in cells with silenced *HMGA2* [171] (Figure 3B). These data suggest that *HOXA9* and *HMGA2* both play an important role in the differentiation process in myeloid cells and that *HOXA9* could be a new target for *HMGA2* regulation [171]. Given the known ability of *HMGA2* to modulate the expression of genes through direct interactions with DNA or by interacting with different transcription factors, it is possible that *HMGA2* can influence the expression of *HOXA9* by interacting directly with the promoter of the latter or indirectly through other mediators by inducing a cascade of signals. 

An interesting work has further highlighted the oncogenic role of *HMGA2* in AML but, at the same time, shown that it to be an important modulator of the disease. The study was based on the profiling of miRNAs expression and was conducted in a de novo pediatric AML cohort. It revealed that four miRNAs were downregulated in AML with inv(16), t(8;21), or mixed lineage leukemia (MLL) rearrangements. By inducing the forced expression of one of these miRNA (*miR-9*), a reduction in leukemic growth and the initiation of monocytic differentiation in AML t(8;21) cells was observed, both in vivo and in vitro. The suppressive role of *miR-9* is due to the collaboration with *let-7* in order to inhibit the oncogenic LIN28B/*HMGA2* axis [184]. LIN28B belongs to a group of factors that can induce somatic reprogramming of cells into pluripotent stem cells [185] and is overexpressed in different types of cancer [186]. This protein selectively represses the expression of the *let-7* miRNA [187,188,189,190]. *Let-7* miRNAs are tumor suppressors that perform their function by repressing oncogenes such as *MYC, RAS* [78], and *HMGA2*; many tumors have a downregulation of these miRNAs [191]. The concerted action of *let-7* and *miR-9* allows a greater knockdown effect of *HMGA2*, unlocking the differentiation process and blocking uncontrolled growth. These data may open up the possibility of using *miR-9* regulation as a therapeutic option in AML t(8;21) [184].

Gerritsen et al. subdivided AML patients on the basis of two recurrent genetic alterations, the *RUNX1* mutation and t(8;21), and observed that, as compared to t(8;21), the *RUNX1* mutation is able to initiate a specific transcriptional program that provides an important contribution to leukemic transformation. Among the genes showing a differential expression are *MEIS1* [192], *TCF4* [193], and *HMGA2* [194], already described as upregulated in leukemogenesis and that can potentially contribute to the clinical differences between patients who have the *RUNX1* mutation and those with t(8;21) [195].

The studies that followed have in recent years allowed us to consider *HMGA2* as a promising molecular marker for the diagnosis of AML [172]; moreover, intervening in the balance of different signal pathways, it can be an important modulator of the main cellular mechanisms and, therefore, a potential therapeutic target in the treatment of AML [164,171,172]. A recent study conducted in the Canadian population showed that using a prognostic test based on the expression of *HMGA2* allows a better stratification of AML patients in different risk groups. This can reduce unnecessary treatments and can allow early scheduling of patients in first remission for allogeneic transplant, improving AML patients’ leukemia-free survival and OS. All this not only leads to economic savings but also above all facilitates better patient management and disease outcomes [196].

## 5. HMGAs Involvement in Lymphoid Malignancies

Acute lymphoblastic leukemia (ALL) is the main cause of cancer death in children and adults and the most frequent lymphoid malignancy in children, with B-cell ALL (B-ALL) accounting for 25% of all childhood cancers [197]. T-cell ALL (T-ALL), instead, accounts for 15% of all ALL and is a very aggressive leukemia featuring the rapid accumulation of T lymphocytes precursors, with a higher risk of relapse as compared to B-ALL [198]. Unluckily, existing stratification risk criteria are narrow and are based primarily on clinical characteristics and underlying genetic lesions [199]. Unlike B-lineage ALL, no risk stratification exists for T-ALL, but a few possible prognostic factors do. Many recurrent genetic alterations have been discovered as oncogene activators on the basis of mutation or overexpression [200]; one example is the activation of *NOTCH1* mutations, identified in almost 50% of human T-ALLs [201]. Considering the number of relapses, clarifying the molecular basis of refractory disease is an urgent need in order to promptly recognize patients destined to relapse and to schedule them for more intensive chemotherapeutic treatments.

Several studies examined the HMGAs’ role in lymphoid disease pathogenesis. 

Both in the mouse transgenic model and in human T- and B-ALLs, *HMGA1a* is expressed at levels two- to 10-fold above the normal T- and B- cells [46]. Misexpression of the human *HMGA1*b isoform in mouse led to the development of a T-cell leukemia with a natural killer (NK) phenotype [47]. Indeed, mice with a complete absence of *HMGA1* showed decreased numbers of T-cells and B-cell and myeloid expansion and in some cases developed B-cell leukemic transformation and B-cell lymphomas [202]. In these mice, spleen tissues revealed an increased expression of the Recombination Activating Gene 1/2 (RAG1/2) endonuclease that might be responsible for the high rate of abnormal *IGH* rearrangements observed in these neoplasias [202]. 

As regards the interaction of HMGA proteins with NOTCH1 signaling, studies on mouse thymic lymphomas detected both the overexpression of HMGA1 and the activation of NOTCH1 signaling [203]. *HMGA1* is a downstream target of Notch1 signaling, as shown by the presence of two NOTCH1/RBPJ cobinding sites of T/CTCCCACA in *HMGA1* promoter regions and the consequent direct regulation of *HMGA1* transcription by NOTCH1. Indeed, in human T leukemia cells, knockdown of HMGA1 provoked a significantly impaired cell growth and decreased expression of cyclin D and cyclin E. The observation of complexes between HMGA1 and retinoblastoma (RB) protein indicated the involvement of HMGA1 in cell cycle regulation and proliferation through NOTCH1 signaling, corroborating its leading role in leukemogenesis [203].

T-ALL frequently shows deletion within the *CDKN2A/2B* tumor suppressor locus that has been found in more than 70% of tested T-ALL samples [204]. As transgenic mice overexpressing *HMGA1* develop aggressive T-ALL [205], the group by Di Cello et al. aimed to understand whether *CDKN2A/2B* genomic loss cooperates with *HMGA1* in T-ALL development [65]. They reported that complete loss of *CDKN2A* accelerated leukemogenesis in *HMGA1* transgenic mice that showed marked splenomegaly and significantly decreased OS compared with *HMGA1* transgenic/*CDKN2A* WT mice. *HMGA1a* transgenic/*CDKN2A* null leukemias presented a T-cell origin immunophenotype with the expression of Th y 1.2, CD3, CD8, and α β T-Cell Receptor (TCR); a markedly accelerated development; and disease recapitulating the salient clinical and pathologic features of human T-ALL. Exploration of the publicly available databases showed that *HMGA1* is one of the genes most frequently overexpressed in pediatric T-ALL compared with control BM samples, supporting the speculation that *HMGA1* could play a causative role in T-ALL and could thus be a potential therapeutic target [65]. 

A recent study explored the correlation of *HMGA1* expression with relapse in pediatric ALL [206]. Investigation of gene expression profiles generated from leukemic blasts from 86 children with relapsed B-ALL showed a significant increase in the mean expression of *HMGA1* at relapse. Further consultation of published databases evidenced *HMGA1* among the top 11% of genes as compared to B-cell progenitors in a prior study of pediatric B-ALL [207] and in the top 21% of overexpressed genes as compared to normal BM in another pediatric B-ALL study [208]. In this latter study, a high *HMGA1* expression was noted in B-ALL samples, with a translocation in *cMYC Proto-Oncogene* (*cMYC)* t(8;14). *HMGA1* is a cMYC transcriptional target [62], and *HMGA1* high expression is maybe linked to *cMYC* translocations. As *HMGA1* has a pivotal role in ES cells and cellular reprogramming [209] and has been identified as a leukemic stem cell signature in a murine model of AML [30], it could trigger refractory disease by inducing stem-like transcriptional programs [206]. 

A study on T-ALL with t(5;14)(q35;q32.2) highlights the involvement of HMGA1 in the mechanism of dysregulation of *TLX3* or *NKX2-5* homeobox genes as a consequence of 5q35 juxtaposition with 14q32.2 breakpoints dispersed across the *BCL11B* downstream genomic desert [210]. In vitro experiments with DNA inhibitory treatment in t(5;14) cells showed the presence of enhancers with multiple regulatory stigmata about 1 Mbp downstream of *BCL11B*. Further knockdown of either HMGA1 or PU.1 inhibited ectopic *NKX2-5* and *TLX3* expression, highlighting their implication in leukemic gene deregulation in t(5;14), likely through the generation of a putative t(5;14) enhanceosome in which HMGA1 is bound near enhancers, and PU.1 to promoters of homeobox genes [210]. As t(5;14) T-ALL is a major clinical entity, this information could be important to identify potential therapeutic targets such as DNA inhibitory treatments and histone deacetylase inhibition [210].

The HMGA1-STAT3 pathway also seems to play a crucial role in lymphoid neoplasias, such as T-ALL and Burkitts lymphoma, an aggressive lymphoid disease involving more mature B cells. HMGA1 regulates the expression of various tumor progression driver genes, such as *STAT3*, a signal transducer and activator of transcription [58,211] for which expression is associated to a refractory status in diverse tumors [212]. HMGA1 induces *STAT3* expression that has a main role in inflammation, malignant transformation, and tumor progression [3,58,213,214]; STAT3 overexpression also leads to hematologic malignancies [3,58]. In vitro blocking of STAT3 function induces apoptosis in T-ALL cells derived from a *HMGA1* transgenic model [58]. Indeed, the inhibition of HMGA1 or STAT3 function prevents Burkitt leukemia cells colony formation [58]. Blocking STAT3 function could be a new therapeutic approach in T-cell lymphoid tumors driven, at least in part, by HMGA1 altered expression, considering that HMGA1 has been demonstrated to drive leukemic transformation and refractory disease by inducing STAT3 and a stem-like transcriptional network [211] (Figure 4A). 

As regards *HMGA2*, a cytogenetic analysis of abnormalities at the 12q12-q14 chromosomal locus conducted in 78 adult B- and T-cell ALL cases showed that, in 26% of cases, a submicroscopic deletion was present at the 12q14.3 locus targeting the region containing the *HMGA2* gene [215]. 

Indeed, *HMGA2* physiological function can also be indirectly altered. miRNA *let-7*b is frequently downregulated in *KMT2A*-rearranged ALL as a consequence of DNA hypermethylation of its promoter region, similar to what happens with the *CDKN2A* gene [216,217]. As a consequence, *let-7b*-regulated target genes are upregulated, such as *HMGA2*, that is normally negatively regulated by *let-7*b [74,79]. In 2015, Wu et al. demonstrated that, in infant leukemic cells, particularly in those positive to *KMT2A-AFF1 (MLL-AF4)*, there was upregulation of *HMGA2*. As HMGA2 is a negative regulator of *CDKN2A* gene, a series of in vitro studies on *KMT2A-AFF1*-expressing cell lines showed that the use of the HMGA2 inhibitor netropsin in addition to the demethylating agent 5-azacytidine upregulated and sustained the expression of *CDKN2A*, resulting in a higher growth suppression as compared to treatment with 5-azacytidine alone (Figure 4B). These data highlighted that the *let-7*b-HMGA2-CDKN2A axis regulates cell proliferation of leukaemic cells and could be a possible molecular target for the treatment of infant ALL with *KMT2A-AFF1* [218].

Supported by the findings regarding the key role of HMGA proteins in several important lymphoid pathways, it seems reasonable to suppose that, in the next years, HMGAs may be used as prognostic and/or targetable markers, even though further studies will be needed to corroborate the current evidence and to fully exploit them in clinical practice.

## 6. HMGA Proteins as Targetable Markers

In view of their role in carcinogenesis, HMGA proteins could be suitable candidates for molecular therapy against HMGA-overexpressing cancers. As they bind AT-specific minor grooves of DNA, their binding could be inhibited by other DNA minor groove binders/ligands, such as distamycin and netropsin [20,219], that have been studied as potential new therapies for several types of human cancer [2,220].

The search for HMGA1 inhibitors is ongoing. Despite toxicity issues that still need to be addressed, promising preliminary data have been obtained for flavopiridol and FR900482 [4,221].

A series of studies explored the use of STAT3 as a targetable marker in aggressive lymphoid malignancies overexpressing HMGA1. Nanoparticle delivery of STAT3 with G-quartet oligodeoxynucleotides that specifically inhibit STAT3 binding to DNA has been shown to have antileukemia effects in an HMGA1 transgenic model of aggressive T-ALL [211]. Indeed, BP-1-102, a salicylic acid-based inhibitor, was tested in preclinical models of ALL and Burkitts lymphoma with STAT3 activation [212], demonstrating that BP-1-102 inhibits STAT3 phosphorylation at tyrosine 705, preventing STAT3 dimerization, DNA binding, and the activation of downstream genes. It also decreases nuclear and cytoplasmic phosphorylated STAT3. Further treatment of cultured cells derived from B-lymphoid tumors previously shown to express high levels of HMGA1 and STAT3, including B-ALL, Burkitts leukemia, and T-ALL cells, showed a decreased proliferation [222]. However, there was no antitumor efficacy in murine xenograft models; the reason for this is still uncertain and may lie either in the limited exposure to the drug or in the induction of oncogenic pathways alternative to the one inhibited by BP-1-102 [222]. These experiments also helped to show that BP-1-102 treatment represses HMGA1 [222], highlighting that not only does HMGA1 induce STAT3 expression [58] but even STAT3 feeds forward to upregulate HMGA1, leading to an enhanced expression of both genes during tumor progression. In fact, a putative STAT3 consensus DNA binding site (TTN5AA) was found in the *HMGA1* promoter region; it is conserved in humans and mice and could mediate STAT3 dimer binding and transactivation. In conjunction with direct transactivation, STAT3 could induce downstream factors that upregulate *HMGA1* expression [222].

Furthermore, Bortezomib (BTZ) also seems to influence HMGA1 levels. This is a proteasome inhibitor used to suppress Diffuse large B-cell lymphoma (DLBCL) progression [223,224,225]. DLBCL is the most frequent non-Hodgkin lymphoma, showing variable clinical presentations as well as variable therapy responses [226,227,228]. In vitro studies on DLBCL CRL-2630 cells showed that BTZ treatment significantly inhibited the proliferation of DLBCL CRL-2630 cells [229]. After studies showing that exposure to BTZ induced an upregulation of *miRNA 198* (*miR-198*) while depletion of the miR significantly reversed the inhibitory effect of BTZ on cell proliferation [230], further studies on *miR-198* revealed the 3′-UTR of *HMGA1* as a downstream target of *miR-198*, suppressing the expression of *HMGA1* in DLBCL cells. This evidence links HMGA1 to BTZ treatment that decreased the level of HMAG1 and inhibited the migration of DLBCL cells [229,231,232,233].

Indeed, a recent screening of about 36,000 compounds indicated a class of phosphodiesterase inhibitors that help to suppress *let-7* targets [234]. These potential drugs augment cAMP levels and elevate *let-7* levels, inhibiting *let-7* target genes such as *HMGA2* and *MYC* and decreasing growth in multiple cancer cell lines [234].

In the treatment of CML patients, the introduction of specific tyrosine kinase inhibitors (TKI) has promoted great progress in patient management [235,236]. Several TKIs are currently enrolled in common practice (such as imatinib, nilotinib, and dasatinib), the use of which is strictly related to adverse events and to the observation of resistance phenomena developing during therapy [237,238]. These issues increase the need to devise alternative therapies; in recent years, studies on the use of epigenetic therapy associated with other therapies have seemed to offer valid therapeutic tools [239]. In this regard, the Vitkeviciene team investigated the potential role of two epigenetic modulators, EGCG (epigallocatechin-3-gallate) and BIX-01294 (1-benzylpiperidin-4-yl)-6,7-dimethoxy-2-(4-methyl-1,4-diazepan-1-yl) quinazolin-4-amine) in two AML and CML cell lines [240].

EGCG is a naturally occurring catechin in green tea. This molecule is known to have many functions, including antibacterial, antioxidant, and anti-inflammatory actions, but it also seems to have a potential antineoplastic role, inhibiting proliferation and inducing apoptosis in different tumors [241,242]. BIX-01294 is a synthetic molecule that allows the specific inhibition of EHMT2/G9a histone methyltransferase. EHMT2/G9a is an important protagonist of gene silencing mechanisms [243,244]. A study in vitro showed that EGCG and BIX-01294, acting as epigenetic modulators, cause cell cycle arrest in the G0/G1 phase; the lines treated with EGCG also showed an increased level of ATM, HMGA2, phosphorylation of ATM, and Senescence-associated (SA)-β-galactosidase staining, for which concerted action favors the cellular senescence phenomenon [240]. Although apoptosis as compared to the induction of senescence is believed to be more effective in the treatment of cancer, the use of substances capable of inducing senescence offers a great advantage in terms of therapy planning [245].

## 7. Conclusions

The studies conducted so far characterize HMGA proteins as master regulators of tumor progression, refractory disease, and cancer stem cell properties, providing strong evidence about their crucial function in malignant hematopoiesis.

The goal in the coming years could be to discover whether and how best this knowledge could be exploited in routine procedures now that high throughput genomic and transcriptomic technologies are available. 

Indeed, the significance of a tailored therapy against tumors overexpressing HMGAs needs to be established in order to understand whether the differential expression of these proteins could be used in personalized therapeutic strategies, since several specific drugs for HMGA-involved pathways are available and others are being tested. Because HMGA proteins control gene transcription by remodeling chromatin and recruiting transcription factor complexes, an alternative approach to block their function could be by targeting downstream pathways induced by HMGAs. The differential function and involvement of HMGA proteins in tumor and normal cells could give such targeted therapy an advantage in terms of specificity and low toxicity.

Finally, the finding of HMGA1 involvement in reprogramming of somatic cells to fully pluripotent cells paves the way for its use in regenerative medicine to reprogram normal cells to stem-like cells [209].

Today, all these issues remain topics open to close research in this field, which will certainly need to continue, bearing in mind the potential of these proteins as personalized medicine targets.

## Figures and Tables

**Figure 1 cancers-12-01456-f001:**
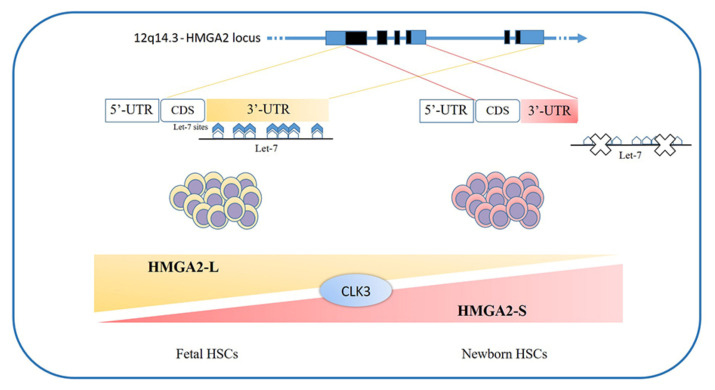
*HMGA2* development-specific isoforms: HSCs feature distinct alternative splicing patterns in various key hematopoietic regulators, one of which is HMGA2. In humans, two alternative splicing isoforms have recently been discovered: the *HMGA2-L* isoform, which is sensitive to degradation by *let-7* and is dominant in fetal HSCs, and the *HMGA2-S* isoform, that is resistant to degradation by *let-7* and is prevalent in newborn HSCs. The two isoforms reveal alternative 3’UTR usage; they share the first three exons and differ in their terminal exon usage (including C-terminal domains and 3’UTRs). The *HMGA2-S* 3’UTR is one third the length of the *HMGA2-L* 3’UTR and lacks the seven *let-7* sites as well as the other most conserved miRNA sites. The balance between alternative isoforms and the escape from miRNA-mediated control seems to depend on the activity of the splicing kinase CLK3, which promotes exon skipping and thus influences the *HMGA2* splicing pattern [8]. HSCs-hematopoietic stem cells; 3’UTR-3′-Untranslated Region.

**Figure 2 cancers-12-01456-f002:**
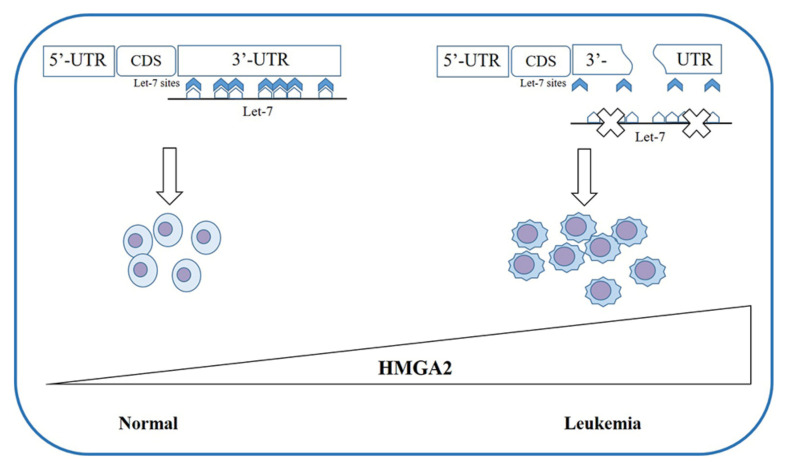
Different expression of *HMGA2* in hematological diseases: Following the truncation of the 3′-UTR, the binding sites for *let-7* are lost and the negative regulation of miRNA on the expression of *HMGA2* no longer occurs. Overexpression of *HMGA2* leads to a clonal growth advantage of the hematopoietic cell, as observed in MPN, PNH, and MDS patients. HMGA2-High mobility group AT-Hook 2; 3′-UTR-3′-Untranslated Region; MPN-Myeloproliferative neoplasms; PNH-paroxysmal nocturnal hemoglobinuria; MDS-myelodyslastic syndrome.

**Figure 3 cancers-12-01456-f003:**
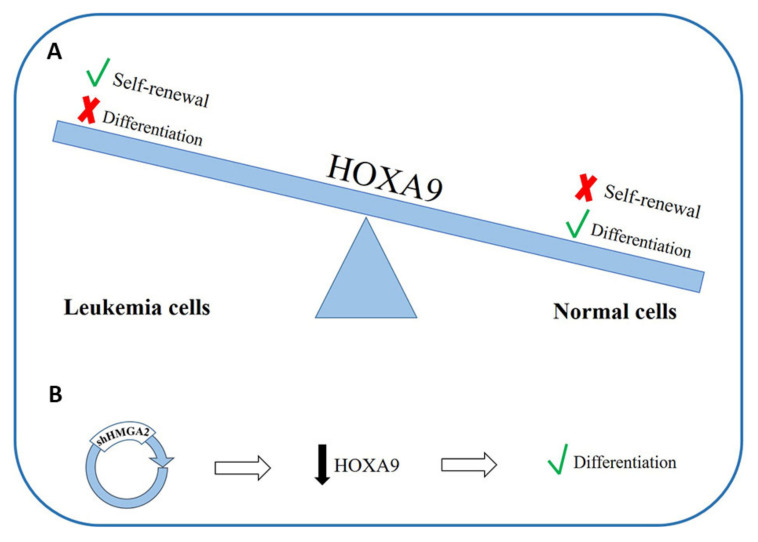
The role of *HOXA9* regulation in normal myeloid differentiation: (**A**) *HOXA9* overexpression maintains the self-renewal capacity of leukemic stem cells and blocks the differentiation process. (**B**) In some leukemia cell lines, HMGA2 knockdown affects *HOXA9* expression, and this influences the differentiation processes in myeloid cells. HOXA9-Homeobox A9; HMGA2-High mobility group AT-Hook 2.

**Figure 4 cancers-12-01456-f004:**
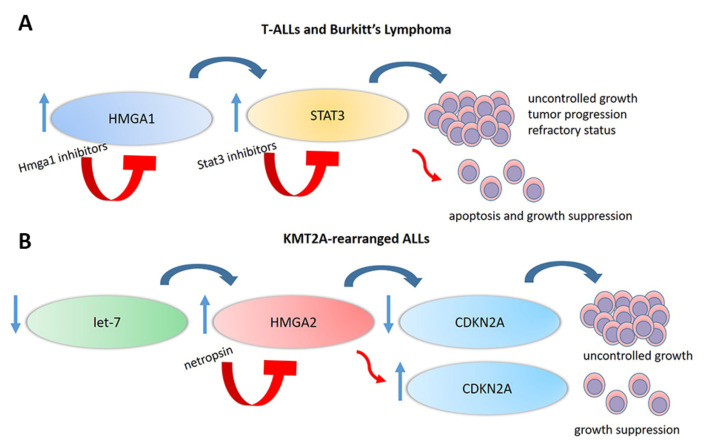
HMGA involvement in lymphoid malignancies: Graph representing some HMGA-controlled pathways involved in lymphoid tumorigenesis. (**A**) The HMGA1–STAT3 pathway plays a crucial role in lymphoid neoplasias, such as T-ALL and Burkitts lymphoma. HMGA1 induces *STAT3* expression [58,211] that has a main role in malignant transformation and tumor progression and in determining a treatment refractory status [212]. In vitro blocking of HMGA1 or STAT3 function seems to be a promising new therapeutic approach in T-cell lymphoid tumors to induce apoptosis and to inhibit cell growth [211]. (**B**) *KMT2A*-rearranged ALLs frequently show miRNA *let-7*b downregulation [216,217]. As a consequence, *let-7*b-regulated target genes are upregulated, including *HMGA2*. *HMGA2* overexpression has an impact on its downstream targets, such as the *CDKN2A* gene, that is negatively regulated by HMGA2. This circumstance triggers leukemic proliferation. The use of the HMGA2 inhibitor netropsin in addition to the demethylating agent 5-azacytidine has been demonstrated to upregulate and sustain the expression of *CDKN2A*, resulting in a higher growth suppression compared to treatment with 5-azacytidine alone [218]. HMGA-High mobility group AT-Hook; T-ALL-T-cell Acute lymphoblastic leukemia.

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
