# Peer review of "HMGA Proteins in Hematological Malignancies"

_cancers, 2020, doi:10.3390/cancers12061456_

Round 1
Reviewer 1 Report
General comments:
The topic of the review is interesting and should be very useful for researchers in the field. But, several part of the manuscript must be largely rewritten in order to be reduce so that a take home message can be easily taken. This rewriting preclude the publishing of the review in its present form. Specific comments could help the authors in their rewriting.
Specific comments:
All along the manuscript: space error: lines 67, 137, 151, 154, 168, 213, 215, 235, 238, 243, 244, 254, 256, 257, 286, 329, 331, 346, 414, 444, 651
Spelling mistake
Reference: lines 391, 400
Line 137 : « LIN28- and HMGA2-induced » Is this correct?
Line 231 « …JAK2 mRNA and phosphorylation of of STAT3 and …» remove one «of»
Line 330 « The etiology of the disease is very complex and there are many risk factors. The etiology of the disease is very complex and there are many risk factors. Several studies have …» remove the sentence repeated
- HMGA Proteins in Oncology
Line117 « this confirms the presence of functional sites ….» Are the authors sure that functional studies were conducted? Otherwise the authors should use «suggest»
- HMGA Proteins in Hematopoiesis
Line 135 « HMGA2 is thus upregulated in fetal HSCs, and LIN28- and HMGA2-induced overexpression increased the self-renewal activity of adult HSCs in transplanted irradiated hosts. » The idea here is not clear
Line 139 « ….; however, HMGA2 overexpression in adult HSCs did not mimic the effects of elevated LIN28B, that triggers a fetal lymphoid differentiation program. » The link between HMGA2 overexpression in adult HSCs and the effects of elevated LIN28B is not clear
Line 148 Why HMGA2 isofoms are not introduce in the introduction part as HMGA1 isoforms,
- HMGAs Involvement in Myeloid Malignancies
Line 260 «The overexpression of HMGA2 associated with the 3'-UTR deletion was found not only in MPN patients, but also in those with MDS and MDS/MPN [86,128–132], and also in two patients with paroxysmal nocturnal hemoglobinuria (PNH)(Figure 2). » It is indicate 2 patients. From which study they come from? Is it necessary to indicate this information?
Line 263 « PNH is characterized by the absence on then cell surface of the glycosyl phosphatidylinositol protein on the stem cell. All cell lines deriving from this stem cell show the same alteration and an increased expression of HMGA2 due to the truncation of the 3'-UTR [89] » Are functional studies sustaining this idea? If yes please mention it.
Lines 292-297: Could the authors clarify this part
Line 367: Are functional studies sustaining this idea? If yes please mention it.
Line 368 to 370 Could the authors simplify the message
Line 376 to 379 Could the authors better express the idea
- HMGAs Involvement in Lymphoid Malignancies
This part must be shorten in order to highlight relevant data
Conclusion
Could the authors rewrite some part of the conclusion in order to clarify the take home message?
Author Response
- “All along the manuscript: space error: lines 67, 137, 151, 154, 168, 213, 215, 235, 238, 243, 244, 254, 256, 257, 286, 329, 331, 346, 414, 444, 651”
All space errors have been corrected as suggested.
- “Spelling mistake Reference: lines 391, 400”
Corrected.
- “Line 137: « LIN28- and HMGA2-induced » Is this correct?”
Yes, it is. It stands for ‘LIN28-induced and HMGA2-induced overexpression…’.
- “Line 231 « …JAK2 mRNA and phosphorylation of of STAT3 and …» remove one «of»”
Done.
- “Line 330 « The etiology of the disease is very complex and there are many risk factors. The etiology of the disease is very complex and there are many risk factors. Several studies have …» remove the sentence repeated”
Done.
- “Line117 « this confirms the presence of functional sites ….» Are the authors sure that functional studies were conducted? Otherwise the authors should use «suggest»”
The sentence has been modified, as suggested. (L116)
- “Line 135 « HMGA2 is thus upregulated in fetal HSCs, and LIN28- and HMGA2-induced overexpression increased the self-renewal activity of adult HSCs in transplanted irradiated hosts. » The idea here is not clear”
The sentence has been rewritten. (L134-138)
- “Line 139 « ….; however, HMGA2 overexpression in adult HSCs did not mimic the effects of elevated LIN28B, that triggers a fetal lymphoid differentiation program. » The link between HMGA2 overexpression in adult HSCs and the effects of elevated LIN28B is not clear”
The concept has been clarified. (L138-140)
- “Line 148 Why HMGA2 isofoms are not introduced in the introduction part as HMGA1 isoforms”
As suggested, HMGA2 isoforms have been introduced in the introduction part. (L35-38)
- “Line 260 «The overexpression of HMGA2 associated with the 3'-UTR deletion was found not only in MPN patients, but also in those with MDS and MDS/MPN [86,128–132], and also in two patients with paroxysmal nocturnal hemoglobinuria (PNH)(Figure 2). » It is indicate 2 patients. From which study they come from? Is it necessary to indicate this information?”
The study has been mentioned, as requested. (L277-279)
- “Line 263 « PNH is characterized by the absence on then cell surface of the glycosyl phosphatidylinositol protein on the stem cell. All cell lines deriving from this stem cell show the same alteration and an increased expression of HMGA2 due to the truncation of the 3'-UTR [89] » Are functional studies sustaining this idea? If yes please mention it.”
The study has been mentioned, as requested. (L279-282)
- “Lines 292-297: Could the authors clarify this part”
The sentence has been rewritten. (L306-312)
- “Line 367: Are functional studies sustaining this idea? If yes please mention it.”
The study has been mentioned, as requested. (L372-378)
- “Line 368 to 370 Could the authors simplify the message”
The sentence has been rewritten. (L379-382)
- “Line 376 to 379 Could the authors better express the idea”
The sentence has been rewritten. (L387-390)
- “HMGAs Involvement in Lymphoid Malignancies: This part must be shorten in order to highlight relevant data”
As suggested ‘HMGAs involvement in Lymphoid Malignancies’ part has been shortened, indeed some concepts have been moved to ‘HMGA Proteins in Hematopoiesis’ part.
- “Conclusion, Could the authors rewrite some part of the conclusion in order to clarify the take home message?”
The conclusion part has been rewritten, as suggested.
Reviewer 2 Report
This is a comprehensive review of HMGA proteins in hematopoiesis and hematologic malignancies.
Suggestions:
- Introduction is quite long, would suggest distilling down the critical information into a more succinct section
- In the MPN section some paragraphs only have one sentence, some of these could be combined.
- Line 306 - sentence does not make sense. The article cited is about MPN, not MDS and PNH.
- Lines 337-338 doesn't make much sense as a standalone paragraph. If this sentence is kept into the manuscript additional explanation would be helpful.
Author Response
- “Introduction is quite long, would suggest distilling down the critical information into a more succinct section”
The introduction part has been shortened, as suggested.
- “In the MPN section some paragraphs only have one sentence, some of these could be combined.”
The paragraphs have been combined to allow a smoother reading, as suggested.
- “Line 306 - sentence does not make sense. The article cited is about MPN, not MDS and PNH.”
The sentence has been rewritten to better understand the quote. (L319-322)
- “Lines 337-338 doesn't make much sense as a standalone paragraph. If this sentence is kept into the manuscript additional explanation would be helpful.”
The sentence has been integrated into the paragraph to improve understanding. (L349-351)
Round 2
Reviewer 1 Report
The authors took into account the reviewer’ comments, but there are still some mistakes
HMGA Proteins in Hematopoiesis
Lines 130 -134: Could the authors rewrite the folowing sentence “In 2013, in mouse models, Copley et al. demonstrated that HSCs developmentally ……potential was reduced [109].”
Line 134 sentence similar to lines 89 and 90 Could the authors clarify the idea
Line 199: Maybe I am wrong but “Indeed” should be removed.
HMGAs Involvement in Myeloid Malignancies
Line 249: remove one “of “ “and phosphorylation of of STAT3 and AKT”
Line 306-308 could the authors rewrite the sentence
Line 344- 346 Could the authors remove the repeated sentences This point not corrected “The etiology of the disease is very complex and there are many risk factors. The etiology of the disease is very complex and there are many risk factors”
Author Response
- “Lines 130 -134: Could the authors rewrite the folowing sentence “In 2013, in mouse models, Copley et al. demonstrated that HSCs developmentally ……potential was reduced [109].”
The sentence has been rewritten, as requested. (L107-110)
- “Line 134 sentence similar to lines 89 and 90 Could the authors clarify the idea”
The idea has been clarified as requested. (L110-115)
- “Line 199: Maybe I am wrong but “Indeed” should be removed.”
Done (L176)
- “Line 249: remove one “of “ “and phosphorylation of of STAT3 and AKT””
Done.
- “Line 306-308 could the authors rewrite the sentence”
As requested, the sentence has been rewritten. (L282-284)
- “Line 344- 346 Could the authors remove the repeated sentences This point not corrected “The etiology of the disease is very complex and there are many risk factors. The etiology of the disease is very complex and there are many risk factors”
The repeated sentence has been removed, as requested. (L320-321)